# HPA Axis Responsiveness Associates with Central Serotonin Transporter Availability in Human Obesity and Non-Obesity Controls

**DOI:** 10.3390/brainsci12111430

**Published:** 2022-10-25

**Authors:** Christian Schinke, Michael Rullmann, Julia Luthardt, Mandy Drabe, Elisa Preller, Georg A. Becker, Marianne Patt, Ralf Regenthal, Franziska Zientek, Osama Sabri, Florian Then Bergh, Swen Hesse

**Affiliations:** 1Klinik und Hochschulambulanz für Neurologie, Charité-Universitätsmedizin Berlin, Freie Universität Berlin and Humboldt-Universität zu Berlin, Charitéplatz 1, 10117 Berlin, Germany; 2Berlin Institute of Health at Charité, Universitätsmedizin Berlin, Anna-Louisa-Karsch Straße 2, 10178 Berlin, Germany; 3Department of Neurology, University of Leipzig, 04109 Leipzig, Germany; 4Department of Nuclear Medicine, University of Leipzig, 04109 Leipzig, Germany; 5Integrated Research and Treatment Center (IFB) Adiposity Diseases, University of Leipzig, 04109 Leipzig, Germany; 6Rudolf-Boehm-Institute of Pharmacology and Toxicology, Clinical Pharmacology, Faculty of Medicine, University of Leipzig, 04109 Leipzig, Germany; 7Translational Centre for Regenerative Medicine, University of Leipzig, 04109 Leipzig, Germany

**Keywords:** obesity, hypothalamic–pituitary–adrenal axis (HPA), serotonin transporter (5-HTT), positron emission tomography (PET), reward

## Abstract

Background: Alterations of hypothalamic–pituitary–adrenal (HPA) axis activity and serotonergic signaling are implicated in the pathogenesis of human obesity and may contribute to its metabolic and mental complications. The association of these systems has not been investigated in human obesity. Objective: To investigate the relation of HPA responsiveness and serotonin transporter (5-HTT) availability in otherwise healthy individuals with obesity class II or III (OB) compared to non-obesity controls (NO). Study participants: Twenty-eight OB (21 females; age 36.6 ± 10.6 years; body mass index (BMI) 41.2 ± 5.1 kg/m^2^) were compared to 12 healthy NO (8 females; age 35.8 ± 7.4 years; BMI 22.4 ± 2.3 kg/m^2^), matched for age and sex. Methods: HPA axis responsiveness was investigated using the combined dexamethasone/corticotropin-releasing hormone (dex/CRH) test, and curve indicators were derived for cortisol and adrenocorticotropic hormone (ACTH). The 5-HTT selective tracer [^11^C]DASB was applied, and parametric images of the binding potentials (BP_ND_) were calculated using the multilinear reference tissue model and evaluated by atlas-based volume of interest (VOI) analysis. The self-questionnaires of behavioral inhibition system/behavioral activation system (BIS/BAS) with subscales drive, fun-seeking and reward were assessed. Results: OB showed significant positive correlations of ACTH curve parameters with overall 5-HTT BP_ND_ (ACTH_AUC_: *r* = 0.39, *p* = 0.04) and 5-HTT BP_ND_ of the caudate nucleus (ACTH_AUC_: *r* = 0.54, *p* = 0.003). In NO, cortisol indicators correlated significantly with BP_ND_ in the hippocampus (cortisol_AUC_: *r* = 0.59, *p* = 0.04). In OB, BAS reward was inversely associated with the ACTH_AUC_ (*r* = −0.49, *p =* 0.009). Conclusion: The present study supports a serotonergic-neuroendocrine association, which regionally differs between OB and NO. In OB, areas processing emotion and reward seem to be in-volved. The finding of a serotonergic HPA correlation may have implications for other diseases with dysregulated stress axis responsiveness, and for potential pharmacologic interven-tions.

## 1. Introduction

Obesity is a leading cause of preventable disease [1] and frequently associated with the metabolic syndrome as a cluster of cardiovascular risk factors [2,3] as well as psychosocial distress [4], stigmatization [5] and psychiatric illness [6]. While the pathogenesis of obesity is heterogeneous, including genetic, epigenetic and environmental factors [7], alterations of the stress response and central serotonergic signaling are acknowledged contributors in the pathogenesis of obesity [8,9,10] and may contribute to some of its unfavorable physical and mental health conditions ity [11,12]. Stress hormones are profoundly intertwined with the intake, distribution and expenditure of energy [13]. Dysregulation of the hypothalamic–pituitary–adrenal (HPA) axis is associated with obesity [11], which is most prominently recognized in Cushing’s syndrome [14]. Experimentally, chronic mild stress exposure leads to hyperphagia with preference for highly palatable food [15] as well as anhedonia in disease models of depression [16], which parallels the dose-dependent relation of chronic psychosocial stress with the prevalence of the metabolic syndrome [17] or depressive symptoms in humans [18,19]. The preference for high caloric food in individuals with an enhanced stress-induced cortisol reactivity [20,21] implicates vulnerability of this neuroendocrinological endophenotype in the pathogenesis of obesity.

HPA axis activity is modulated by serotonin and vice versa [22,23]. Its depletion experimentally increases [24,25], while serotonin administration decreases food intake [26]. In humans, the appetite-decreasing effect of serotonin has been exploited by the application of fenfluramine or sibutramine as enhancers of serotonin (5-hydroxytryptamine, 5-HT) concentrations in the synaptic cleft [27], or lorcaserin as an 5-HT_2c_ agonist [28]. The 5-HT transporter is a critical modulator of serotonergic activity since it limits concentrations of 5-HT in the synaptic cleft by its reuptake into the presynaptic neuron to terminate its action [29]. Changes in serotonergic signaling have been linked to obesity [30,31] and reward sensitivity in OB [32]. An interaction of HPA axis activity with the central serotonin system could be shown in animal models of stress-related disorders [33], in genetic association studies in humans [34], in treatment trials of major depression [35], as well as imaging studies of central serotonin transporter or receptor availability with HPA axis reactivity in stress-related disorders [36,37]. The association between HPA axis activity and the serotonergic system is of importance, as both systems are involved in the pathogenesis of obesity and perhaps in some of its metabolic and mental complications, but constitute potentially pharmacologically modifiable targets [38,39].

There is evidence that both serotonergic signaling and neuroendocrine factors are closely related to the stress response, which plays a role in the development of obesity. To directly investigate a potential association of the serotonergic system with the HPA axis in this context, we applied the combined dexamethasone/CRH (dex/CRH) test for HPA axis reactivity [40,41] and PET imaging with carbon-11 labelled *N*,*N*-dimethyl-2-(2-amino-4-cyanophenylthio)benzyl-amine ([^11^C]DASB [31]) in OB compared with NO. We further explored whether HPA activity is related to 5-HTTLPR genotype and reward seeking behavior. We hypothesized that HPA axis activity and 5-HTT availability are associated, and that these relations may regionally differ between OB and NO. We expected that HPA axis responsiveness is negatively associated with reward sensitivity in OB.

## 2. Methods

### 2.1. Study Population

Twenty-eight otherwise healthy individuals with class II (BMI 35.0–39.9 kg/m^2^) or class III obesity (≥40.0 kg/m^2^) were recruited from the outpatient clinic of the Integrated Research and Treatment Centre, Adiposity Diseases, a university clinic for obesity and associated disorders (OB, 21 females; age 36.6 ± 10.6 years; body mass index (BMI) 41.2 ± 5.1 kg/m^2^). Twelve NO (8 females; age 35.8 ± SD 7.4 years; BMI 22.4 ± 2.3 kg/m^2^) from the local community were recruited by advertisements in the local media and matched for age and sex. All participants were free of psychiatric or neurological diseases, vascular encephalopathy, head trauma, drug or alcohol misuse, pregnancy or breast-feeding. No participant had used centrally acting medications, illicit drugs or glucocorticoid treatment for at least 6 months. Participants received a general physical examination along with neurological status and were seen by an experienced psychiatrist conducting a semi-structured interview, and only participants without symptoms or signs of clinically relevant depression were included. For depressive symptoms, participants were further screened by the Beck Depression Inventory II (BDI-II) [42]. In the OB group, three individuals scored >14, which is the cut-off value for mild depression (two individuals with a score of 15 and 1 with 17). We abstained from excluding these individuals, as the BDI sores contrasted with the clinical impression of the investigating psychiatrist, who did not find signs of clinically relevant depression, and as scores seemed to be primarily driven by somatic complaints rather than from depression. The consumption of alcohol and nicotine was recorded for both groups.

Routine laboratory investigations and urine screening were performed. Five additional individuals were studied but subsequently excluded from the analysis due to (i) insufficient PET data statistics, (ii) severe psycho-social distress on the day of the dex/CRH test, (iii) suspected Cushing’s disease based on the dex/CRH test results, (iv) reported severe alcohol abuse in a second interview and (v) reported tachycardia after dexamethasone ingestion so the dex/CRH test was not completed. Written informed consent was obtained from all individuals. The study was conducted in accordance with the Declaration of Helsinki and approved by the ethics committee of the Medical Faculty of the University of Leipzig registered under the number 206-10-08032010 and by the German Bundesamt für Strahlenschutz/Federal Office for Radiation Protection (number Z5-22461-2-2011-002). The study was registered in the European clinical trial database (EudraCT 2012-000568-32) and the German Clinical Trials Register (DRKS S00003537).

### 2.2. Procedures

MR imaging: Structural MR images were acquired using a 3T Siemens scanner and a T1-weighted 3D magnetization prepared rapid gradient echo (MP-RAGE) sequence (time of repetition 2300 ms, time of echo 2.98 ms, 176 slices, field of view (FoV) 256 × 240 mm, voxel size 1 × 1 × 1 mm) for PET-MRI co-registration and (with other sequences based on the Alzheimer’s Disease Neuroimaging Initiative protocol) for exclusion of brain pathologies such as diffuse or confluent white matter hyperintensities in T2-weighted images, tumors, stroke but not malformation without functional impairment.

PET imaging: [^11^C]DASB was synthesized according to [43]. Dynamic PET was performed for 90 min after intravenous bolus injection (90 s) of (mean) 484 ± 10 MBq [^11^C]DASB using the ECAT EXACT HR+ scanner (Siemens, Erlangen, Germany, intrinsic resolution at the center: 4.3 mm, axial resolution: 5–6 mm field of view 15.5 cm, 3–4 mm full width at half maximum) in three-dimension acquisition mode. Emission scan acquired 23 frames (4 × 0.25, 4 × 1, 5 × 2, 5 × 5, 5 × 10 min). We used a 10 min transmission scan (from 3 68Ge sources), which was performed before the emission scan, for attenuation correction and iterative reconstruction (10 iterations, 16 subsets) in transverse image series (63 slices, 128 × 128 matrix, voxel size 2.6 × 2.6 × 2.4 mm^3^) with a Hann filter (cut-off 4.9 mm) for image reconstruction. Parametric images of 5-HTT binding potential (BP_ND_) were generated from the PET data by the multi-linear reference tissue model with two parameters (MRTM2) and the cerebellar cortex as the reference tissue [31,44].

Imaging analysis: Regional analyses of BP_ND_ values were performed after co-registration of BP_ND_ images with individual 3D magnetic resonance imaging (MRI) data using PMOD software (Version 3.4) for re-alignment and stereo-tactical normalization (according to the anterior commissure-posterior commissure line), as well as for delineating the volumes of interest (VOI). Selected VOIs included brain regions that are involved in appetite regulation, stress axis regulation, emotion or behavior [31,41]: frontal cortex (FC), orbitofrontal cortex (OFC), dorsolateral prefrontal cortex (dlPFC), anterior cingulate cortex (ACC), insula, hippocampus, amygdala, nucleus accumbens (NAcc), head of the caudate, putamen, thalamus, hypothalamus, substantia nigra (SN) and ventral tegmental area (VTA), dorsal raphe nucleus (DRN), midbrain, pons.

Dex/CRH test: The dex/CRH test was performed as described previously [10,40]. In brief, participants received 1.5 mg dexamethasone per os at 23.00 h on the day before CRH application. Study participants were advised to come in a relaxed state, avoiding psychological or physical stress exceeding their daily routine. On the day of the test, an intravenous cannula was inserted into the cubital vein at 14.30 h and kept patent by isotonic saline infusion at a 20 mL/h rate. The first blood sample (before CRH stimulation) was taken at 15.00 h. At 15.02 h, an i.v. bolus of 100 µg of synthetic human CRH (Ferring, Kiel, Germany) was applied. Subsequent blood samples were taken at 15.30 h, 15.45 h, 16.00 h, and 16.15 h. Samples were stored at 4 °C and centrifuged immediately after the test; serum and plasma, respectively, were taken off, and samples were stored at −80 °C until assayed. Cortisol was measured in serum; ACTH concentrations in EDTA plasma. Dex/CRH test and 5-HTT PET imaging were conducted with a median time difference of 27 days (IQR 5.5–60.5 d).

Assay methodology. Commercial chemiluminescence immunoassays were used to measure hormone concentrations as described previously [41]. ACTH concentrations were determined with Liaison^®^ ACTH, DiaSorin, Italy, and cortisol concentrations with Cobas^®^, Roche Diagnostics, Germany, following the manufacturers’ instructions. Respective intra- and inter-assay coefficients of variation (CV) for ACTH were below 7.7% for a target value of 9.53 pmol/L and below 7.3% for a target value of 62.3 pmol/L. Representative intra- and interassay CVs for cortisol were below 3.2% for a target value of 86.2 nmol/L and below 2.0% for a target value of 1120 nmol/L. The functional sensitivity of 20% CV was set to be 0.84 pmol/L for ACTH and 8.5 nmol/L for cortisol, according to the manufacturer’s instruction.

#### 5-HTTLPR Genotyping

Bi-allelic status was determined as described previously [32]. In brief, genomic DNA was extracted from 1 mL of a 5–10 mL peripheral blood sample with EDTA as anticoagulant. Isolation steps were performed by applying pegGold DNA Mini kit (pegLab, Erlangen, Germany) according to the manufacturer’s instructions. 5-HTTLPR gene polymorphism was determined with a standardized polymerase chain reaction amplification procedure [45]. The primer sequences used were 5′-GAGGGACTGAGCTGGACAAC-3′ and 5′-GCAGCAGACAACTGTGTTCATC-3′, with a product length of ~ 620 bp for the L-allele and 583 bp for the S-allele. Primers were purchased from Invitrogen (Paisley, UK).

### 2.3. Questionnaires Assessing the Behavioral Inhibition System (BIS)/Behavioral Activation System (BAS), Depression and Anxiety

The sensitivity to reward and punishment was assessed using the behavioral inhibition system/behavioral activation system (BIS/BAS) self-questionnaires as described previously to assess the responsiveness of BAS and BIS personality characteristics (reactivity of the aversive motivational system and appetitive motivational system) with the three subscales drive, fun-seeking and reward [32,46]. Depressive symptoms were measured using the BDI-II [42]. BDI scores were measured on the dex/CRH test day, BIS/BAS reward scores and SCL-90 within the first 4 weeks after participant inclusion.

Statistical analysis: SPSS 25 was used for statistical analysis. Graphs were created with GraphPad Prism 8 (La Jolla, CA, USA). For statistical analysis of dex/CRH test results, post-CRH concentrations for ACTH and cortisol (30 min after CRH application), maximum concentration (MAX) and area under the time course curve above zero according to the trapezoid rule (“ground” area-under-the-curve; AUC) were calculated from the plasma hormone concentrations measured (five time points mentioned, see Figure 1A,B). ACTH/cortisol ratios were calculated for each indicator. Normal distribution was tested using the Shapiro–Wilk test, which yielded *p* < 0.05 for all neuroendocrine data. After exclusion of asymmetries of corresponding brain regions, BP_ND_ was averaged side-by-side to reduce the number of variables and multiple comparisons. Relationships between BP_ND_ and dex/CRH test parameters, i.e., ACTH and cortisol MAX and AUC, respectively, were analyzed using Spearman rank correlation for categorical data. All data are given as median with interquartile range or mean ± standard deviation (SD), according to data distribution and unless otherwise stated. Two-tailed significance was applied. The Mann–Whitney *U* test (not-normally distributed data) or unpaired t-test (data with normal distribution) were conducted for group comparison (2 groups) or Kruskal–Wallis test with Dunn’s correction (>2 groups, data not-normally distributed or small sample size). Results were considered significant at *p* < 0.05.

## 3. Results

### 3.1. Study Population Characteristics

Participant characteristics and epidemiological data are summarized in Table 1. The study population referenced here was included in the studies of Schinke et al. [10] and Hesse et al. [31]. OB had higher BDI scores than NO (OB 6.5 [3,4,5,6,7,8,9,10,11] vs. NO 0 [0–3.3], *p <* 0.0001), without reaching subthreshold of mild depression. BAS fun was slightly lower in OB than in NO (OB 11.1 ± 2.0 vs. NO 12.4 ± 1.7, *p* = 0.028, see Table 1).

### 3.2. Individuals with Obesity Tend to Have a Higher HPA Axis Responsiveness and a Higher Adrenal Sensitivity to ACTH

Parameters of HPA axis responsiveness of OB vs. NO are described in detail in [10]. Individuals with obesity showed higher HPA axis responsiveness than their non-obese counterparts in the dex/CRH test with statistical significance for cortisol 30 min after stimulation with CRH (see Table 2 and Figure 1A,B), as well as a higher adrenal sensitivity to ACTH as measured by a lower ACTH/cortisol ratio (see Table 2). ACTH related parameters did not substantially differ between OB and NO (Table 2). Explorative analyses according to participants’ sex showed no significant differences in the ACTH or cortisol response between female and male NO or female and male OB (Appendix A). In male individuals with obesity, ACTH measured in the post-CRH sample was higher than in male non-obesity controls (OB: 1.95 pmol/l vs. NO: 1.05 pmol/l, *p* = 0.024, Kruskal–Wallis test with Dunn’s multiple comparison correction, see Appendix A).

### 3.3. HPA Axis Responsiveness Differentially Relates to 5-HTT Availability between OB and NO

Regions of interest delineated for 5-HTT BP_ND_ are shown in a representative image in Figure 1C. In OB, there was a significant positive correlation of ACTH_AUC_ with overall 5-HTT BP_ND_ (*r* = 0.39, *p* = 0.04), whereas no association was found for cortisol curve parameters (cortisol_AUC_: *r* = 0.09, *p =* 0.65, see Table 2, Figure 2A,B). Region-specific analyses revealed a significant positive relation between ACTH_AUC_ and 5-HTT BP_ND_ of the caudate nucleus (ACTH_AUC_: *r* = 0.54, *p* = 0.003), but not of other brain regions (Figure 3C–F). In NO, no correlation of HPA axis curve parameters with overall 5-HTT BP_ND_ was found (ACTH_AUC_: *r* = 0.05, *p =* 0.88; cortisol_AUC_: *r* = 0.13, *p =* 0.68; see Table 3, Figure 3A,B). Region-specific analyses showed that cortisol but not ACTH curve parameters correlated significantly with 5-HTT BP_ND_ of the hippocampus (*r* = 0.59, *p =* 0.04, see Table 2, Figure 3F).

### 3.4. BAS Reward Scores Relate Differentially to HPA Responsiveness in OB vs. NO

In OB, BAS reward was negatively associated with the ACTH_AUC_ (*r* = −0.49, *p =* 0.009), while the other questionnaires (BDI, SCL-90 anxiety, BAS Drive/Fun, BIS) were not associated with HPA responsiveness (Table 4, Figure 4A,B). In NO, only SCL-90 anxiety scores were linked with cortisol_AUC_ (*r* = 0.58, *p =* 0.049) with borderline significance, but not with the other questionnaires or endocrine parameters (see Table 4).

### 3.5. In Obesity, BAS Reward Scores Are Associated with Overall 5-HTT BP_ND_ and Caudate Nucleus 5-HTT BP_ND_

We further explored if in OB, overall and caudate nucleus 5-HTT BP_ND_ relate to the questionnaire results. We found significant negative associations between overall BAS reward scores and overall 5-HTT BP_ND_ (*r* = −0.57, *p =* 0.002), and 5-HTT BP_ND_ in the head of the caudate (*r* = −0.58, *p =* 0.001), see Table 5 and Figure 4C,D.

### 3.6. Explorative Analyses of HPA Axis Responsiveness according to 5-HTTLPR Genotype Does Not Point towards Substantial Differences between the S/S, S/L and L/L Allele Carriers

The obesity group was divided into three different subgroups according to their 5-HTTLPR-genotype (S/S, S/L, L/L). Two OB had an S/S, 13 OB S/L and 13 OB L/L alleles, whose dex/CRH test results were compared. There was no statistically significant difference in HPA axis responsiveness between individuals with S/L and L/L genotype (for ACTH_AUC_: S/L 7.21 [4.96–10.47] vs. L/L 6.58 [5.40–8.29], *p* ≥ 0.99; for cortisol_AUC_: S/L 258.8 [157.2–806.8] vs. L/L 138.0 [96.5–487.1], *p =* 0.73, Kruskal–Wallis test with Dunn’s correction, respectively). HPA axis responsiveness in S/S allele carriers was in the range of the aforementioned, albeit the small subgroup size only allowed preliminary comparison (S/S ACTH_AUC_: 4.56 [4.10–5.02], *p =* 0.23 [S/S vs. L/S], *p* = 0.40 [S/S vs. L/L]; cortisol_AUC_: 147.8 [119.1–176.6], *p =* 0.96 [S/S vs. L/S], *p* ≥ 0.99 [S/S vs. L/L], Kruskal–Wallis test with Dunn’s correction, respectively), see Figure 5.

## 4. Discussion

The current study adds to the previous notion of an increased HPA axis responsiveness in people with obesity [10], as it associates the neuroendocrine stress response with serotonergic activity in the living human brain. In obesity, ACTH curve indicators were positively associated with averaged 5-HTT BP_ND_ throughout the brain, while region-specific correlations could be found between ACTH curve indicators and 5-HTT BP_ND_ of the caudate nucleus, but not for cortisol parameters. In non-obesity controls, cortisol AUC correlated positively with hippocampal 5-HTT BP_ND_. These findings support a potential serotonergic–HPA relation that regionally differs between OB and NO. These associations may be explained by either (i) a modulatory effect of the serotonergic tone on the HPA axis, (ii) a bottom-up effect of stress hormones on 5-HTT availability, or (iii) a common, unidentified cause, or a co-variate that mediates this relation, respectively.

Central serotonergic neurons project from the raphe nuclei to virtually all brain regions [47]. Serotonin can both facilitate and inhibit HPA activity and vice versa, probably depending, i.a., on the site of action [23,48]. On a hypothalamic level, serotonin agonists experimentally stimulate CRH-containing PVN neurons, subsequently augmenting ACTH and cortisol release [22], which corroborates the acute effect of serotonin reuptake inhibitors on HPA axis activity in humans [49]. Serotonergic signaling from other brain regions, e.g., the prefrontal cortex or the anterior cingulate cortex, was rather suggested to inhibit HPA activity [9,50,51]. Inversely, cortisol increases 5-HT reuptake in vitro [52,53], in vivo [54] and, as measured by the cortisol awakening response (CAR), associates with prefrontal 5-HTT in humans [50]. Interestingly, in OB, ACTH but not cortisol parameters correlated with 5-HTT, which may be explained by an enhanced glucocorticoid clearance in OB [55,56].

While we previously found a negative association of NAT availability in the hypothalamus with HPA axis activity in individuals with obesity [41], the serotonergic-neuroendocrine association of the current study was rather found in the limbic system. Notably, while in NO the HPA response is associated with 5-HTT availability in the hippocampus as an acknowledged glucocorticoid feedback site inhibiting HPA axis activity [57,58], this association localizes to the caudate nucleus in OB, a brain area rather associated with motor function, emotional behavior and reward processing [59]. The hippocampus is known to inhibit HPA axis activity via glutamatergic projections activating GABAergic neurons in the hypothalamic PVN [58], while the role of the caudate nucleus in the context of HPA function is less established. The absence of a direct anatomical or functional connection from the caudate to the hypothalamus [60,61] raises the question of whether another co-variate mediates this relation, such as mood or reward. Only two studies investigated HPA axis responsiveness with in vivo 5-HTT availability by means of [^11^C]DASB, either in healthy volunteers applying the cortisol awakening response (CAR) [50] or in patients with major depression or anxiety disease using the dexamethasone CRH test [36]. While the former showed a positive relation between prefrontal 5-HTT and CAR, suggesting prefrontal serotonergic inhibition of HPA responsiveness [50], the latter found a negative association between thalamic 5-HTT with dex/CRH test parameters and anxiety, supporting that HPA axis dysregulation partly accounts for the effects of altered serotonergic neurotransmission on anxiety [36].

Although the reward questionnaires BAS and BIS did not show any substantial differences between the NO and OB group, it is of note that in OB, BAS reward questionnaires correlated negatively both with HPA responsiveness and with caudate nucleus 5-HTT. The relation of BMI and BAS reward has been extensively studied in individuals with obesity and non-obesity controls, proving rather a non-linear, inverted U-shape relation [62]. In obese participants, a higher BMI was associated with lower BAS reward scores, consistent with a reward deficiency potentially predisposing to hedonic eating [46]. Previously, in depressed individuals, a lower reward dependency, i.e., a lower response to rewarding stimuli, was associated with an impaired suppression of the cortisol response [63]. The finding of an inverse relation between HPA responsiveness and reward sensitivity in OB of our cohort hence fosters the assumption that higher HPA reactors in participants with obesity rather need higher external stimuli to perceive reward, potentially explaining the tendency to rather consume highly palatable food. Of note, caudate nucleus hypofunction in the reward circuitry was previously found to predispose to overeating [59,64,65]. As caudate 5-HTT and HPA axis responsiveness were associated in our OB sample, both being inversely associated with reward sensitivity, it can be assumed that HPA dysregulation partly explains the association between 5-HTT and reward.

OB patients had higher BDI scores than NO without reaching subthreshold depression or its clinical symptoms, and scores did not correlate with HPA responsiveness. The BDI is the most widely used questionnaire to assess depressive symptoms in OB [66] in whom depression is a frequent mental co-morbidity [67]. It is of note that physical symptoms of obesity overlap with symptoms of depression such as altered body scheme, fatiguability or somatic preoccupation, which seem to have driven the higher scores and may explain the lack of clinical signs of depression [66] and the absent association with the endocrine challenge. While HPA axis hyperresponsiveness is a robust finding in patients with depression [40], symptom severity is usually not associated with endocrine parameters, which corroborates our finding [63]. Interestingly, in NO, greater anxiety scores associate with higher HPA responsiveness and are in line with previous findings in patients with affective disorders [36].

Comparative analyses of HPA activity grouped by the 5-HTTLPR genotype in our cohort could only be explorative, as the sample size for the S/S genotype was low with 2/28 individuals, corroborating the low prevalence in the European population [68]. While the role of 5-HTT polymorphisms mediating gene environment interactions in affective disorders is debated [69,70,71], pointing towards only modest effects if existent at all [72], some studies found hints for a higher HPA responsiveness using the dex/CRH test in individuals with depression carrying the S/S allele [36,73] or in healthy S/S females by means of the cortisol awakening response [74], or saliva after a defined psychosocial stressor [75]. Data from our small subgroup of non-depressed OB S/S carriers do not point in this direction. A possible explanation is that although the S/S allele is associated with a lower 5-HTT expression in vitro [76], this association is less pronounced in the living human brain due to, e.g., epigenetic signatures such as gene promotor methylation that are more closely associated with 5-HTT protein expression [32] and further modulate long-term cortisol concentrations [77].

The combination of quantitative 5-HTT PET imaging with endocrine stimulation tests and psychometric measures in the same individuals is a strength of the study. Some potential limitations must be addressed: All individuals received the same dose of dexamethasone and CRH despite varying BMIs, as neuroendocrine testing was performed according to the standard protocol [40,78]. Sufficient suppression by 1.5 mg oral dexamethasone in both groups was indicated by equal pre-CRH concentrations of ACTH and cortisol. A previous dose–response study observed incomplete suppression of ACTH and cortisol only at very low dosages of dexamethasone; the application of 1 mg dexamethasone, which was found to be a near-maximum dose, resulted in equal suppression in overweight and normal weight controls [79]. Serum dexamethasone concentrations do not depend on BMI [80], supporting that dexamethasone measurement does not improve performance of the dexamethasone suppression test at doses of 1 mg or higher. Although clinically relevant depression was an exclusion criterion, three individuals in the OB group had a BDI-II score of 15 or higher, which is the threshold for mild depression. Physical complaints are highly prevalent in OB and overlap with symptoms of depression, complicating its assessment in OB [81]. Although BDI-II scores in the OB group seemed to be primarily driven by items of physical symptoms, we cannot entirely exclude the presence of subclinical depression in these individuals, which may have influenced endocrine or 5-HTT measurement. Despite the relatively large sample size in the obesity group, the number of participants still appeared too small to reliably differentiate effects of sex or, e.g., handedness on the outcome of endocrine and 5-HTT measures or their association. As explorative analyses of HPA axis responsiveness according to 5-HTTLPR genotype were limited by the small sample size of S/S carriers, tri-allelic genotyping should be considered as an alternative approach for 5-HTTLPR assessment in the future since the L_G_ variant is considered to function more similarly to the S allele than to the L_A_ allele, which would allow pooling of the subgroups [82]. Notably, all participants were recruited from the outpatient clinic of the Integrated Research and Treatment Centre, Adiposity Diseases, meaning that these individuals either sought weight reduction programs or were interested in study participation, which may have affected cohort selection towards individuals motivated for healthier behavior. Due to the cross-sectional nature of the study, we cannot answer the question of causality in either direction. However, previous studies showed that trauma experience early in life leads to a higher cortisol stress reactivity by epigenetic modifications [83], suggesting that environmental factors lead to an acquired, fairly stable change in HPA axis reactivity. On the other hand, the finding of a higher HPA responsiveness in healthy first-degree relatives of patients with affective disorders suggests a vulnerability marker of depression [84].

## 5. Conclusions

In conclusion, the current study supports an association between altered neuroendocrine HPA axis responsiveness and central serotonin signaling in OB predominantly affecting brain reward areas. Since alterations of serotonergic signaling, HPA axis activity and rewarding behavior have also been consistently reported in affective disorders [40,48] as well as in metabolic disease [10,11,41,85], these findings may represent a state predisposing for physical and mental illness. Both HPA responsiveness and 5-HTT availability are potentially modifiable [35,86]. It needs to be clarified in longitudinal studies whether addressing, e.g., a dysregulated HPA axis in OB, leads to weight loss, beneficial changes in reward sensitivity or serotonergic signaling in susceptible individuals.

## Figures and Tables

**Figure 1 brainsci-12-01430-f001:**
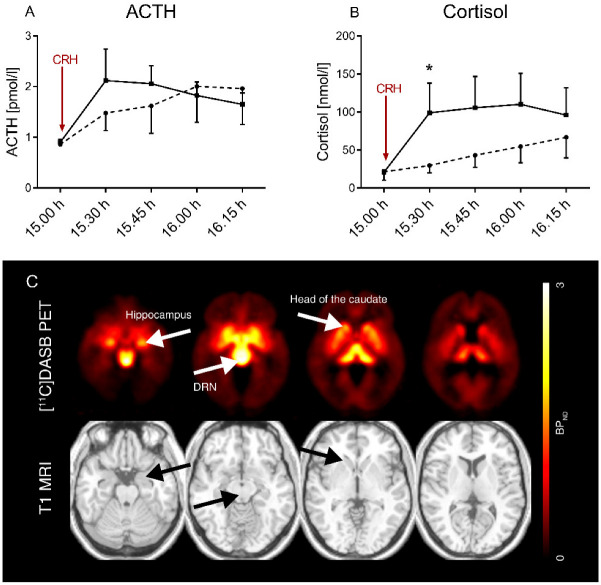
HPA axis responsiveness in the course of time and average parametric [11C]DASB PET imaging. Time course of HPA axis responsiveness by ACTH (**A**) and cortisol response (**B**) to the combined dex/CRH test in individuals with obesity (*n* = 28; squares, solid line) and non-obesity controls (*n* = 12; circles, dashed line). After 1.5 mg dexamethasone, taken orally at 23.00 h the night before testing, 100 µg CRH were applied i.v. at 15.02 h. Post-CRH cortisol (which was 30 min after CRH stimulation, taken at 15.30 h) was significantly higher in OB (marked with *, *p* = 0.01). Data are given as mean with 95% confidence interval. Shown are individuals who also underwent [^11^C]DASB PET; full sample (*n* = 39 vs. *n* = 22 available in [10]). dex, dexamethasone; CRH, corticotropin-releasing hormone; ACTH, adrenocorticotropic hormone. (**C**) Averaged parametric [^11^C]DASB PET image of the entire cohort with a single participant’s T1 MRI (taken from Statistical Parametric Mapping toolbox) in MNI space. DRN: dorsal raphe nuclei.

**Figure 2 brainsci-12-01430-f002:**
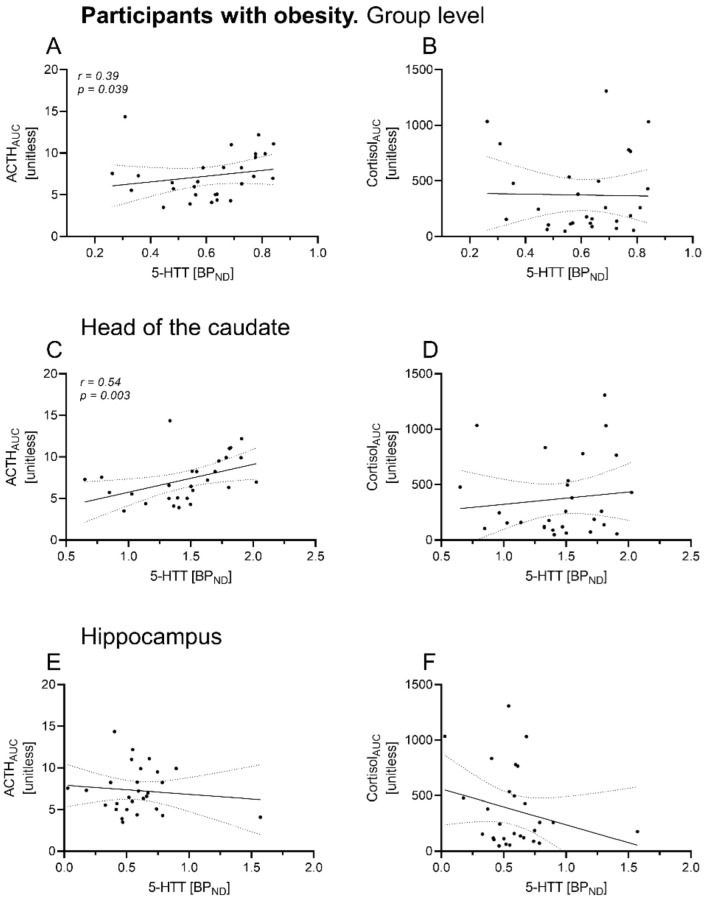
Association of 5-HTT BP_ND_ with HPA axis responsiveness in dedicated brain areas in participants with obesity. Spearman correlation of 5-HTT BP_ND_ of dedicated brain areas with hypothalamic–pituitary–adrenal (HPA) responsiveness in the obesity group. ACTH and cortisol AUC were derived from the dex/CRH test (unitless). Significant positive associations were found for 5-HTT BP_ND_ on group level ((**A**), averaged) and for the caudate nucleus with ACTH_AUC_ (**C**), but not for the hippocampus (**E**). Cortisol AUC did not correlate significantly to 5-HTT binding potential (**B**,**D**,**F**). Dots indicate individual 5-HTT BP_ND_ and HPA axis hormone values of the participants. Data are given with regression line and 95% confidence interval. dex, dexamethasone; CRH, corticotropin-releasing hormone; ACTH, adrenocorticotropic hormone, AUC, area under the curve.

**Figure 3 brainsci-12-01430-f003:**
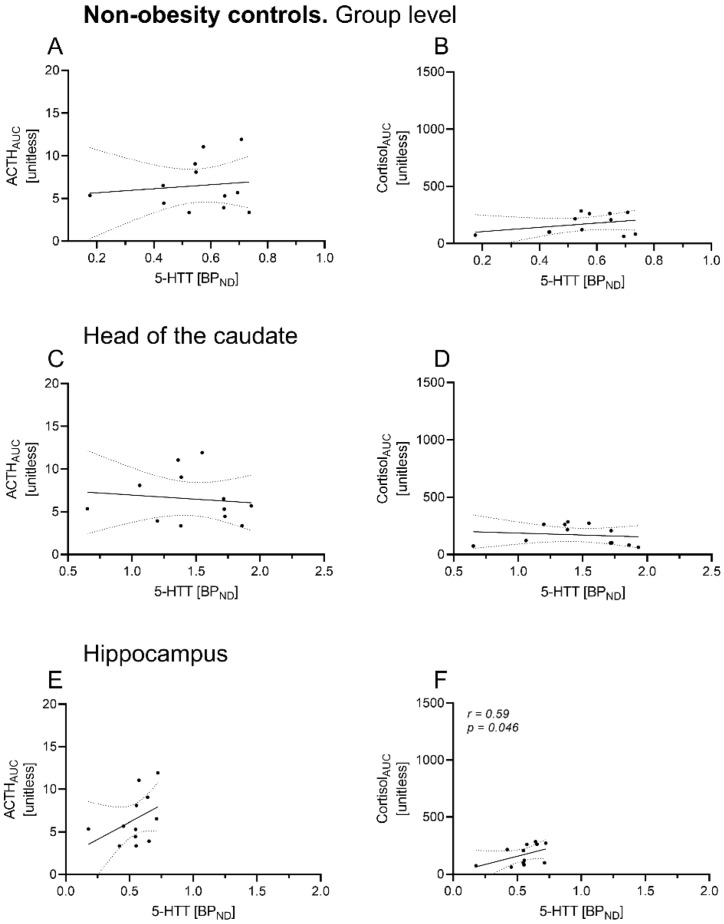
Association of 5-HTT BP_ND_ with HPA axis responsiveness in dedicated brain areas in participants without obesity. Spearman correlation of 5-HTT BP_ND_ of dedicated brain areas with hypothalamic–pituitary–adrenal (HPA) axis responsiveness in non-obesity controls. ACTH and cortisol AUC were derived from the dex/CRH test (unitless). Significant positive association was found for 5-HTT BP_ND_ with cortisol AUC in the hippocampus (**F**), while the other associations did not reach significance (**A**–**E**). Dots indicate individual 5-HTT BP_ND_ and HPA axis hormone values of the participants. Data are given with regression line and 95% confidence interval. dex, dexamethasone; CRH, corticotropin-releasing hormone; ACTH, adrenocorticotropic hormone, AUC, area under the curve.

**Figure 4 brainsci-12-01430-f004:**
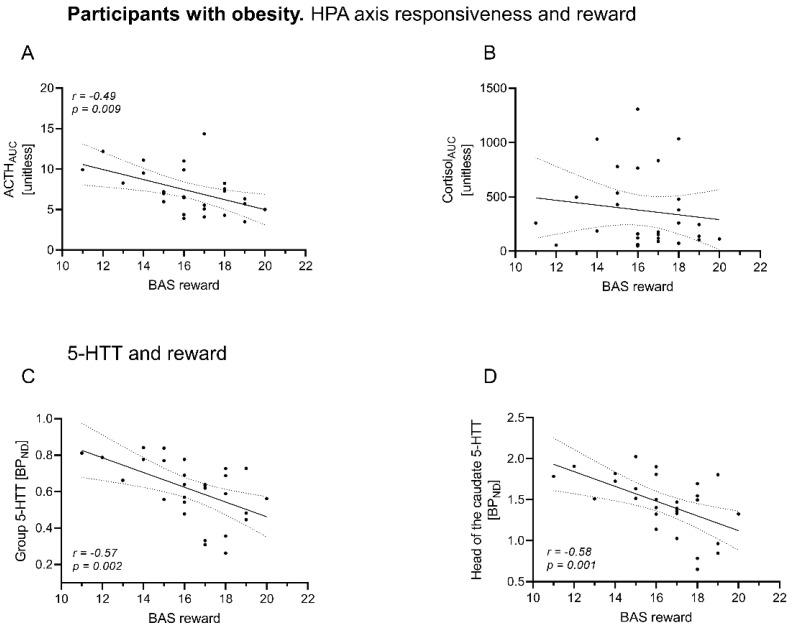
Reward sensitivity in relation to HPA axis responsiveness or 5-HTT BPND of dedicated brain areas in individuals with obesity. Spearman correlation of BAS reward scores with HPA axis responsiveness as measured by (**A**) ACTH and (**B**) cortisol response (AUC) and with averaged 5-HTT BP_ND_ (**C**) and caudate nucleus 5-HTT BP_ND_ (**D**). ACTH and cortisol AUC were derived from the dex/CRH test (unitless). Significant negative associations with BAS reward scores were found for ACTH_AUC_ with BAS reward (**A**) and for 5-HTT BP_ND_ on group level and for the caudate nucleus (**C**,**D**). Data are given with regression line and 95% confidence interval. dex, dexamethasone; CRH, corticotropin-releasing hormone; ACTH, adrenocorticotropic hormone, AUC, area under the curve.

**Figure 5 brainsci-12-01430-f005:**
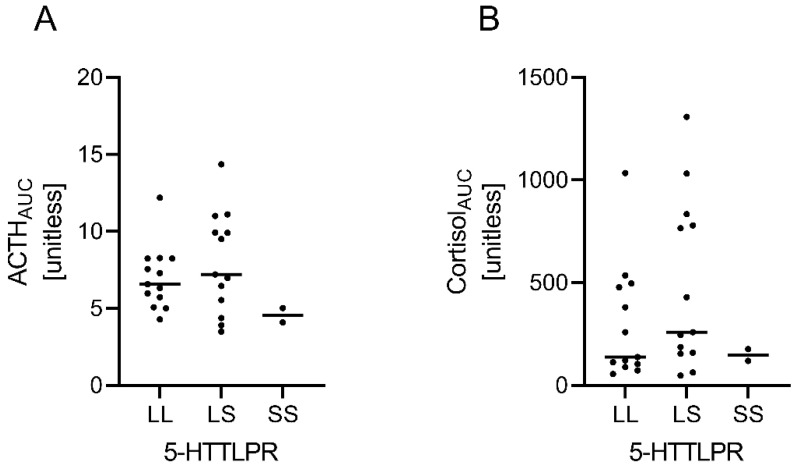
Explorative analyses of HPA axis responsiveness according to 5-HTTLPR genotype in OB. Comparison of the dex/CRH test indicators (**A**) ACTH_AUC_ and (**B**) cortisol_AUC_ (unitless) grouped by 5-HTTLPR genotype. Due to the small subgroup of individuals with the S/S genotype, comparison could only be explorative by nature, and no substantial differences were observed. ACTH, adrenocorticotropic hormone, AUC, area under the curve.

**Table 1 brainsci-12-01430-t001:** Participant characteristics, questionnaires and seasonal data.

	Obesity Group	Non-Obesity Controls	*p*-Value
Number of participants (female)	28	12	
Sex, male/female	7/21	4/8	0.70 ^c^
Age (years)	36.6 ± 10.6	35.8 ± 7.4	0.81 ^a^
BMI (kg/m^2^)	41.2 ± 5.1	22.4 ± 2.3	**<0.0001 ^a^**
Smoking habits, # with score 0/1/2/3	19/0/2/7	10/1/0/1	0.48 ^d^
Beck Depression Inventory	6.5 [3–11]	0 [0–3.3]	**<0.0001 ^b^**
SCL-90-anxiety	49.0 ± 7.6	44.8 ± 5.7	0.09 ^b^
BAS Drive	14 [11.25–14]	13 [12–14]	0.48 ^b^
BAS Fun	11.1 ± 2.0	12.4 ± 1.7	**0.028 ^b^**
BAS Reward	16.3 ± 2.2	17.3 ± 1.9	0.22 ^b^
BIS	19.0 ± 3.8	18.6 ± 2.7	0.85 ^b^
Injected activity (MBq)	481.3 ± 10.9	487.5 ± 6.4	0.08 ^a^

^a^*t*-test; ^b^ Mann–Whitney *U* test; ^c^ Fisher’s exact test; ^d^ Pearson’s chi-square test; BMI, body mass index; Smoking, 0. non-smoker 1. occasionally, 2. not more than 3 cigarettes/d, 3. yes. Data given as median with interquartile range or mean ± standard deviation. **bold:** significant at *p* < 0.05.

**Table 2 brainsci-12-01430-t002:** Dex/CRH test indicators in the obesity group and non-obesity controls.

	Obesity Group (*n* = 28)	Non-Obesity Controls (*n* = 12)	*p*-Value
ACTH_1500h_	<0.84 (<0.84–<0.87)	<0.84 (<0.84–<0.84)	0.42
ACTH_postCRH_	1.66 (1.21–2.42)	1.48 (0.91–1.82)	0.17
ACTH_MAX_	2.10 (1.57–2.98)	1.85 (1.34–3.06)	0.46
ACTH_AUC_	6.78 (5.03–9.21)	5.53 (4.05–8.82)	0.33
Cortisol_1500h_	20.0 (15.2–23.3)	15.8 (11.0–24.2)	0.29
Cortisol_postCRH_	48.9 (32.6–154.1)	25.5 (20.1–38.1)	**0.01**
Cortisol_MAX_	74.4 (38.6–170.5)	66.4 (25.9–100.7)	0.29
Cortisol_AUC_	216.0 (114.2–525.5)	165.4 (87.8–262.5)	0.22
ACTH/cortisol_postCRH_	0.029 (0.013–0.050)	0.059 (0.034–0.097)	**0.02**
ACTH/cortisol_MAX_	0.023 (0.013–0.047)	0.038 (0.021–0.062)	0.22
ACTH/cortisol_AUC_	0.025 (0.014–0.053)	0.043 (0.027–0.066)	0.19

Median (interquartile range). CRH, corticotropin-releasing hormone; ACTH, adrenocorticotropic hormone. MAX, maximum, AUC, area under the curve. Mann–Whitney *U* test was applied for group comparison. ACTH in pmol/l, cortisol in nmol/l, ACTH/cortisol ratio in pmol/nmol; AUC unitless. Data for the whole group of participants (*n* = 39 OB vs. *n* = 22 NO) are available in [10]. **bold:** significant at *p* < 0.05.

**Table 3 brainsci-12-01430-t003:** Spearman correlations of dex/CRH test indicators and 5-HTT BP_ND_ in the obesity group and non-obesity controls.

	Participants with Obesity (*n* = 28)		Non-Obesity Controls (*n* = 12)	
	ACTH_AUC_	Cortisol_AUC_	ACTH_AUC_	Cortisol_AUC_
Group	**0.39 (0.04)**	0.09 (0.65)	0.05 (0.88)	0.13 (0.68)
FC	0.29 (0.14)	0.00 (0.98)	0.23 (0.47)	0.56 (0.06)
OFC/vmPFC	0.35 (0.07)	0.01 (0.96)	0.09 (0.78)	0.08 (0.81)
dlPFC	0.33 (0.09)	−0.11 (0.58)	0.13 (0.68)	0.50 (0.10)
ACC	0.30 (0.12)	0.08 (0.68)	−0.17 (0.60)	0.20 (0.53)
Insula	0.20 (0.30)	0.06 (0.75)	0.23 (0.47)	0.36 (0.25)
Hippocampus	0.05 (0.81)	−0.07 (0.71)	0.55 (0.07)	**0.59 (0.04)**
Amygdala	0.24 (0.22)	−0.08 (0.68)	−0.38 (0.23)	−0.39 (0.21)
NAcc	0.29 (0.13)	0.05 (0.81)	−0.29 (0.37)	0.14 (0.66)
Head of the caudate	**0.54 (0.003)**	0.15 (0.45)	−0.18 (0.57)	−0.34 (0.29)
Putamen	0.24 (0.23)	−0.02 (0.93)	−0.03 (0.93)	−0.11 (0.73)
Thalamus	0.03 (0.89)	−0.18 (0.37)	0.32 (0.31)	0.24 (0.44)
Hypothalamus	0.12 (0.53)	−0.14 (0.49)	0.15 (0.63)	0.17 (0.59)
Substantia nigra/VTA	0.19 (0.34)	−0.22 (0.26)	0.15 (0.65)	0.13 (0.68)
Midbrain	0.22 (0.27)	−0.22 (0.27)	0.16 (0.62)	0.03 (0.91)
Pons	0.04 (0.82)	−0.18 (0.37)	−0.33 (0.30)	0.32 (0.31)
Dorsal raphe nuclei	−0.07 (0.71)	−0.08 (0.70)	0.03 (0.93)	0.14 (0.67)

Spearman correlation and significance (p). MAX, maximum; AUC, area under the curve; ACTH in pmol/L, cortisol in nmol/L; AUC unitless; FC, frontal cortex, OFC/vmPFC, orbito-frontal/ventromedial prefrontal cortex; dlPFC, dorso-lateral prefrontal cortex; ACC, anterior cingulate cortex; VTA, ventral tegmental area; NAcc, nucleus accumbens; **bold:** significant at *p* < 0.05.

**Table 4 brainsci-12-01430-t004:** Spearman correlations of dex/CRH test indicators with questionnaires of depression, anxiety and reward.

	Obesity Group (*n* = 28)		Non-Obesity Controls (*n* = 12)	
	ACTH_AUC_	Cortisol_AUC_	ACTH_AUC_	Cortisol_AUC_
Beck Depression Inventory	0.13 (0.51)	0.13 (0.53)	−0.03 (0.94)	0.13 (0.70)
SCL-90 anxiety	0.06 (0.79)	−0.09 (0.71)	0.31 (0.33)	**0.58 (0.049)**
BAS Drive	−0.19 (0.34)	−0.16 (0.43)	0.16 (0.62)	−0.03 (0.92)
BAS Fun	−0.18 (0.35)	−0.07 (0.72)	0.57 (0.05)	0.35 (0.25)
BAS Reward	**−0.49 (0.009)**	−0.20 (0.32)	0.52 (0.09)	0.04 (0.90)
BIS	−0.25 (0.21)	−0.35 (0.07)	0.27 (0.40)	0.15 (0.65)

Spearman correlation and significance (*p*). MAX, maximum; AUC, area under the curve; BAS, behavioral activation score; BIS, behavioral inhibition score. **bold:** significant at *p* < 0.05.

**Table 5 brainsci-12-01430-t005:** Spearman correlations of questionnaires of depression, anxiety and reward with 5-HTT BP_ND_ of dedicated brain areas in participants with obesity.

	Obesity Group (*n* = 28)					
	Beck Depression Inventory	SCL-90 Anxiety	BAS Drive	BAS Fun	BAS Reward	BIS
Group	0.34 (0.07)	−0.29 (0.21)	−0.26 (0.19)	−0.19 (0.32)	**−0.57 (0.002)**	−0.22 (0.26)
Head of the caudate	0.31 (0.11)	−0.21 (0.37)	−0.11 (0.58)	−0.15 (0.44)	**−0.58 (0.001)**	−0.26 (0.18)

Spearman correlation and significance (*p*). MAX, maximum; AUC, area under the curve; BAS, behavioral activation score; BIS, behavioral inhibition score. **bold:** significant at *p* < 0.05.

## Data Availability

Open data publishing guidelines were followed. Data with description of analyses and calculations are available in the Appendix A.

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
