# Peer review of "HPA Axis Responsiveness Associates with Central Serotonin Transporter Availability in Human Obesity and Non-Obesity Controls"

_brainsci, 2022, doi:10.3390/brainsci12111430_

Round 1

Reviewer 1 Report

Schinke and colleagues sought to examine if there is an association between HPA axis activity and serotonin transporter binding in the brain. The authors used a dexamethasone/corticotropin-releasing hormone (dex/CRH) test in conjunction with [11C]DASB binding and self-reported BIS/BAS measures. The authors state their results support serotonergic-neuroendocrine crosstalk, and that this crosstalk differs in people with obesity. They found people with obesity had increased cortisol 30 minutes after CRH administration and a lower ACTH/cortisol ratio. The authors also found a significant positive association between ACTH and serotonin transporter binding in the caudate nucleus of people with obesity, and a significant positive association between cortisol and serotonin transporter binding in the hippocampus of people without obesity. BAS scores were also negatively associated with ACTH in people with obesity.

The objective of the authors was interesting, and generally the manuscript is well-written.  The combination of serotonin transporter binding with circulating hormone measurements in the same individuals is a particular strength of the study.  Unfortunately, multiple concerns about the methods substantially attenuate enthusiasm for the manuscript.  These reservations are outlined below.  

Major reservations

1.     The doses of dexamethasone and CRH used for the dex/CRH test are of the biggest concern, because interpretations of the dex/CRH test are used in the majority of subsequent analyses performed by the authors.   The authors wrote that participants received 1.5 mg dexamethasone per os and 100 ug CRH iv. With all participants being given the same doses of dexamethasone and CRH regardless of their BMI, despite one group having nearly double the BMI of the other, this generates a troubling confound.  The subsequently observed changes (or lack thereof) in ACTH and cortisol are likely skewed by dramatically different doses per body weight administered to people with obesity and people without obesity.  Accordingly, it is likely that people with obesity received approximately half of the effective doses given to people without obesity.  Therefore, the blunted suppression of ACTH and cortisol responses to CRH following dexamethasone administration are far more likely to do with a basic pharmacologic dose-response effect than to reflect any actual changes in endogenous HPA axis activity resulting specifically from obesity.  To conclude that obesity is in fact directly impacting HPA axis activity, investigators would need to administer body weight-based doses of dexamethasone and CRH, then subsequently observe a sufficiently powered and significant outcome in circulating stress hormone levels.

2.     Another methodological concern is the exclusion criterion for depressive symptoms. The authors used the Beck Depression Inventory, though what version used is unclear. The exclusion criteria used in this paper was 22, which is unusually high compared to both the original scale’s cutoff of 10, and the newer scale’s cutoff of 14. The authors should justify why such a high cut off (22) was used.

3.     Overall justifications and rationales for the authors’ research questions and methodology are lacking.  The introduction and discussion, though mostly (but see #10 below) written well in the grammatical sense, are rather disjointed in logic.  In its current state, the manuscript reads as an attempt to follow up on the authors’ norepinephrine transporter findings with some hastily assembled findings from a parallel investigation.  The writing capabilities of the authors are evident in places though, and they are thus encouraged to invest in revising their manuscript so that readers can clearly follow the logical progression of their experimental queries without having to make individual guesses or perform several internet searches.  Additionally, it will be important for the authors acknowledge known sex differences in HPA responsiveness and obesity, and explain why they did not include sex as a covariate in their analyses.  Best practice would be to include sex as a covariate.  Relevance of the brain regions examined in this study should also be explained.  What was the handedness of participants?  Such information is standard for imaging studies, but missing here.

4.     The authors did not use people first language. Using people first language should be used to respect people with obesity and objectively acknowledge and report on obesity as a condition, not as an identity.

5.     Authors are advised to use more caution in making conclusions based on their data.  The statement “these findings support a serotonergic-HPA crosstalk“ is repeated several times in the manuscript.  However, the study as described is only capable of evaluating an association between serotonin transporter binding and HPA axis activity – it is not possible to determine causality nor interaction with this experimental design.  Further, the manuscript does not describes the temporal order of PET imaging, MRI imaging, dex/CRH testing, and BDI or BIS/BAS scale administration in relation to each other.

6.     Authors should genotype participants for LA and LG variants of L allele (see: Hu X, Oroszi G, Chun J, Smith TL, Goldman D, Schuckit MA. An expanded evaluation of the relationship of four alleles to the level of response to alcohol and the alcoholism risk. Alcohol Clin Exp Res. 2005 Jan;29(1):8-16. doi: 10.1097/01.alc.0000150008.68473.62. PMID: 15654286).  The LG variant is established as functioning more similarly to the S allele than to the LA allele.  Moreover, in the 5-HTTLPR literature it is standard to perform analyses by comparing LA/LA individuals with LG- and S-carriers, particularly as is the case here, where S/S individuals are so few in number.

Minor reservations

7.     Key/legend for Figure 1A/B should be added, and more descriptive headers should placed above graphs in Figures 2-4.  Figure 1C states that PET images are “super-imposed” on MRI images, but in actuality these images are presented separately.

8.     Descriptions in section 2.5 Statistical analysis do not match those in Figure legends.  For example, the former says graphs were mean ± SD, but Figure 1 legend says graphs A and B are mean ± 95% CI.  Also missing are descriptions of how these hormone levels were analyzed  (hopefully with repeated measures?).    

9.     Effect sizes should be reported.

10. Manuscript contains random misspellings and sentence structure/grammatical errors.  An excellent example of such is the first sentence in the introduction.  Another example is the misspelling of “5-hydroxytryptamine”.  “Caudate” is misspelled in Figure 1C, etc.

11. People should be referred to as “participants” and not “subjects” to reflect voluntary participation in the research study.

12. Participants with obesity were all recruited from an outpatient clinic. Why were they at the clinic?  The presence of these participants at an outpatient clinic means they could be inherently different than participants recruited from the community, status of obesity aside. Authors should explain why participants with obesity were only recruited from a single clinic, why they were there, and how this could influence interpretations of the data.

13. In the discussion, the authors reference a “European population”.  Given the authors provide no racial/ethnic data regarding their sample population elsewhere in the manuscript, this statement is distracting.  Further, considering there is no true genetic basis for race, the authors are advised to omit this statement.

Reviewer 2 Report

Dear authors,

I enjoyed reading the manuscript, the work is very well presented, described and discussed. In general, the data is represented in a clear manner and statistics are very well described. It is quite important the crosstalk between serotonergic and neuroendocrine systems in stress and psychiatry and quite novel its relationship with obesity. This study sheds light in key aspects in human obesity on the interaction of both systems which may be relevant to further determining specific mechanisms, biomarkers and future therapies.

I have some minor suggestions that I think it would improve the quality of the current manuscript:

Regarding the figures, I don't totally get the choice of representative graphs. In figure 2 (OB) and 3 (NO) for example refers to table 3 and as I understand it represents group level, head of the caudate and hippocampus based on there were significant differences in one or other groups or parameters. However, in Figure 4 is only represented Obesity group for BAS Reward either for dex/CRH indicators (table 4) or serotonin transporter availability (table 5)* Why not to add also the graph for SCL-90 anxiety since cortisol is also significant in non-obesity controls? 

*(Check also this, table 4 and 5 are different and seem to have the same name and legend).

This study is done in male and female. Both are metabolically different and also have different emotional responses. Have you considered to evaluate if there are gender differences in the parameters evaluated? I am curious and I think that it would be very relevant information. The number of subjects per group is not very big but still enough to find differences or trends. For instance, in the time course HPA responsiveness (F1) the obese seems to have larger error bars meaning higher dispersion and that could potentially reflect two populations. Also in the spearman-correlation graphs it would be better identified if each dot/individual is expressed in two groups of colors representing males and females so one could see the distribution among both genders and such correlations.

OB patients had higher BDI scores, which is quite interesting given the fact that one of the inclusion criteria was OB but otherwise healthy subjects. Yet they did not reach subthreshold nor clinical symptoms. I think this is a point that could be discussed better in the discussion. Do you think this finding may suggest a higher vulnerability to develop diagnosis of depression at any other stage in life (i.e in chronic stress situation, or trauma) while others being resilient? 

In the discussion a paragraph regarding limitations of the study should be considered.
